# Organizational Health Literacy in the Context of Employee Health: An Expert-Panel-Guided Scoping Review Protocol

**DOI:** 10.3390/ijerph19074381

**Published:** 2022-04-06

**Authors:** Lara Lindert, Lukas Kühn, Paulina Kuper, Kyung-Eun (Anna) Choi

**Affiliations:** Center for Health Services Research, Brandenburg Medical School Theodor Fontane, 16816 Neuruppin, Germany; lukas.kuehn@mhb-fontane.de (L.K.); paulina.kuper@mhb-fontane.de (P.K.); anna.choi@mhb-fontane.de (K.-E.C.)

**Keywords:** employee health, occupational health, corporate health management, workplace health promotion, SARS-CoV-2

## Abstract

Health literacy (HL) is an interplay of individual and organizational health literacy (OHL). While individual HL has been intensively studied, the importance of OHL has become a greater focus of research attention. The National Action Plan Health Literacy in Germany emphasizes the promotion of HL in all areas of everyday life, including occupation and the workplace. The proposed scoping review aims at identifying and evaluating definitions, empirical studies and instruments on OHL targeting employee recipients. The search will be conducted in two consecutive steps and guided by expert-panel discussions in accordance to the method of Consensus Development Panels. The search will be conducted in Web of Science, PubMed and Google Scholar according to the methodological framework of Arksey and O’Malley and supplemented by the snowball principle and a hand search. All records will be included that were published until the final search date. To define eligibility criteria, the PCC framework of the Joanna Briggs Institute is used. The scoping review will critically discuss whether a new definition of OHL in the context of employee health is of purpose for future research and practice. Nonetheless, it will provide orientation in the context of employee health, also facing the consequences of SARS-CoV-2.

## 1. Introduction

Many countries are confronted with insufficient levels of their population’s health literacy (HL) [1]. The recent European Health Literacy Study 2019–2021 (HLS_19_) identified judgment of treatment options, protection against illnesses using mass media information and information finding with regard to mental health problems as the most difficult tasks in general HL. Furthermore, aspects of navigational HL (regarding healthcare reforms, patients’ rights, health insurance coverage), communicative HL (regarding physician consultation time, personal views and preferences), digital HL (regarding reliability of information, commercial interests, solving health problems by using information) and vaccination HL (regarding personal vaccination needs, information on recommended vaccinations) provide hurdles for individuals [2]. Low HL is associated with negative health effects [3,4,5] and an increased risk of death [6]. According to the HLS_19_, education was found to be predictive for Navigational, Communicative and Vaccination HL, while age predicted Digital HL. Furthermore, especially financial deprivation and self-perceived level in society predicted general and specific HL [2].

According to Sørensen, HL “entails people’s knowledge, motivation and competences to access, understand, appraise, and apply health information in order to make judgments and take decisions in everyday life concerning healthcare, disease prevention and health promotion to maintain or improve quality of life during the life course” [7]. To improve population HL, it is essential to focus on skills and abilities of individuals but also to consider the requirements and the complexity of the system they live in [8]. Within the last years, HL has gained more and more scientific and political attention on the national and international levels. The World Health Organization (WHO)/Europe initiated the Action Network on Measuring Population and Organizational Health Literacy (M-POHL) to regularly measure HL. The first survey was carried out in 17 countries (Austria, Belgium, Bulgaria, Czech Republic, Denmark, France, Germany, Hungary, Ireland, Israel, Italy, Norway, Portugal, Russian Federation, Slovakia, Slovenia and Switzerland) in 2019 (HLS_19_) [1,2]. According to the HLS_19_ Consortium of the WHO Action Network M-POHL, “health policy should develop strategies to improve people’s Navigational HL, specifically interventions on systemic and organizational levels to make health systems more health-literate, user-friendly, and easier to navigate” [2].

Organizational health literacy (OHL) “is described as an organization-wide effort to transform organization and delivery of care and services to make it easier for people to navigate, understand, and use information and services to take care of their health” [9]. For example, digital applications open up the opportunity to facilitate access to the health system and to health care services and also to lower time-related usage thresholds [8]. Especially in the continuum between prevention and rehabilitation, HL is of special interest, as HL can contribute to prevent or slow down the deterioration of the disease situation and strengthen health resources [10,11,12,13].

So far, OHL concepts mainly address healthcare organizations and consider patients as recipients of OHL [14,15,16,17,18,19,20,21,22]. However, “placing greater emphasis on heath literacy outside of healthcare settings has the potential to impact on preventative health and reduce pressures on health systems” [7]. The National Action Plan Health Literacy in Germany defines fields of action and recommendations that, i.a., emphasize the promotion of health literacy in all areas of everyday life, including occupation and the workplace [1].

Employees’ HL is also of special interest for employers: HL can positively influence individual skills and work ability and reduce occupational hazards and injuries. Employees’ work ability ensures organizations’ competitiveness and overall productivity [23,24,25,26]. In recent study results, HL was associated with absenteeism, eventually mediated by chronification of diseases. However, results differ between countries, and future research needs to further examine the correlation between HL and absenteeism [2]. The reduction in occupational hazards, injuries and absenteeism are an indication that HL also has an effect on economic costs (e.g., fewer absence days in the company due to incapacity to work). In a scoping review on the costs of limited HL, Eichler et al. [27] state that the costs of limited HL are potentially significant. In 2009, limited HL was assumed to cause 3–5% of the total healthcare budget in Canada [28].

In Germany, employers are subject to the Occupational Health and Safety Act and have a duty to detect and prevent or mitigate potential health hazards for employers. Furthermore, in the event of longer periods of absence, they are also obliged to offer employees support in form of an occupational integration management [29]. It is not only up to employees to take care of health issues, but employers also bear responsibility.

Ehmann et al. [26] addressed HL on individual level of employees and outlined the current state of research on work-related HL. In a holistic view, HL demonstrates an interplay of individual (individual skills and abilities) and organizational (system demands and complexities) HL. Low HL does not exclusively result from a lack of knowledge, lack of motivation or insufficient skills of the individual, but it is also decisively shaped by the societal, lifeworld and social conditions under which individuals live and the demands of life situations and environmental aspects [8,30,31]. Therefore, the aim of this scoping review is to discuss a definition of OHL with regard to employee health (step 1) and subsequently identify empirical studies that focus on OHL to improve employee health (step 2).

Furthermore, due to the SARS-CoV-2 pandemic, employees were required to work from home when possible with both positive (e.g., flexibility) and negative effects (e.g., higher workload, distrust of leaders, less social contacts). The SARS-CoV-2 pandemic could be seen as triggering the beginning of a gradual culture change in organizations [32,33]. However, low earners and employees with low skill levels are less likely to be able to work from home. As employees without home office access bear a disproportionately higher burden of the pandemic (higher home office potential is negatively correlated with both regional reports of short-time work and regional infection figures) [34] and as financial deprivation and self-perceived level in society predict general and specific HL [2], those differences in individuals with and without home office possibilities have to be considered. The results will be discussed with regard to the requirements this entails.

The results of the HLS_19_ indicate that context-specific definitions are necessary when it comes to HL, as different associations can be identified for different HL contexts. It is stated that research on further specific HL is necessary and may contribute to the next wave of measuring HL [2]. We are committed to discussing a definition of OHL in the context of employee health for the following reasons:The definition may especially help employers that are not located in the healthcare sector to focus and work on OHL in the context of employee health.A specific definition will reinforce the relevance of the topic outside the healthcare sector and ideally raise awareness among employers. This could help to relieve the healthcare system.To improve population HL, it is essential to not only focus on the skills and abilities of individuals but also to consider the requirements and the complexity of the system they live in [8]. Different hurdles and opportunities in the various areas of the complex overall system underline the need for a specific definition.

The aim of this study is not to create a completely new definition of OHL but to summarize the current state of research in this specific field and to place it in the overall context of HL to give orientation for policy makers, practitioners (e.g., human resource managers) and researchers. A definition will only be given when existing definitions do not consider all relevant aspects of OHL in the context of employee health that will be revealed within the scoping review. Furthermore, definitions should include measurable outcomes [31]. Against this background, an expert panel will critically discuss whether a “new” definition of OHL in the context of employee health is of purpose for researchers and practitioners or if already existing concepts can easily be adapted to the special context of employee health (step 1). Based on the consensus, a literature search will be conducted to identify relevant empirical studies in this field (step 2).

The scoping review aims at providing orientation for policy makers, researchers and practitioners by giving information on what matters in the context of OHL and employee health and what specifically needs to be considered against the background of changes caused by SARS-CoV-2.

The research question to be answered in step one is the following:(RQ1)What concepts and definitions of organizational health literacy in the scope of employee health can be identified?

The research questions to be answered in step two are the following:
(RQ2)What empirical studies can be identified that focus on (aspects of) organizational health literacy to improve employee health? What concepts and definitions of organizational health literacy are used?(RQ3)What measures are used to operationalize organizational health literacy with regard to employee health?

## 2. Materials and Methods

We have chosen the scoping review as our methodology, as scoping reviews, i.a., aim at creating working definitions or delineating the content boundaries of a topic [35]. The scoping review will be registered on the Open Science Framework database. The study will be conducted by a multidisciplinary team of four researchers (L.L., L.K., P.K., K.-E.C.) with expertise in psychology, clinical physiotherapy, health services research and rehabilitations science. Furthermore, an expert-panel is planned to be set up to discuss study results at different times of the process. The panel will be composed of 5 to 11 experts from different disciplines, as this size was found to be most beneficial across several consensus methods. The panel will be organized in accordance to the method of Consensus Development Panels, which is commonly used in healthcare research [36].

The scoping review will focus on all articles that address OHL in the context of employee health. The whole process will involve two steps.

In the first step, the aim is to identify literature that defines or conceptualizes OHL in the context of employee health. In this step, all types of references and evidence will be considered. The results will then be discussed in the expert-panel, also with regard to SARS-CoV-2. We have deliberately decided for this first step as it is essential: only when criteria are clear is a literature search focusing on empirical studies in this specific field of purpose.

In a second step, eligible empirical studies in several scientific databases will be identified, based on the result in step 1. Again, the results will be discussed in the expert panel, also against the backdrop of the results from the first panel. If reasonable, more expert rounds may be held.

Both steps of the scoping review will be conducted according to the methodological framework of Arksey and O’Malley that comprises the following five stages [35]:

### 2.1. Identifying the Research Question

The research questions that will be addresses in the scoping review are as follows:(RQ1)What concepts and definitions of organizational health literacy in the scope of employee health can be identified?(RQ2)What empirical studies can be identified that focus on organizational health literacy with regard to employee health? What concepts and definitions of organizational health literacy are used within these studies?(RQ3)What measures are used to operationalize organizational health literacy?

### 2.2. Identifying Relevant Studies

To identify relevant studies, we use the PCC framework of the Joanna Briggs Institute [37] to define eligibility criteria (see Table 1).

The search strategy will be guided by eligible criteria. Databases for study selection will be Web of Science, PubMed and Google Scholar. Regarding (RQ1), the results of a hand search will also be reported.

Keywords will be defined according to the characteristics of the PCC framework and extended by related terms. Within domains, keywords will be combined by the Boolean operator “OR”. For the connection of domains, the Boolean operator “AND” will be used. If necessary, the Boolean operator “NOT” will be used to exclude irrelevant results. Keywords will be truncated if applicable and listed in a table with related terms in rows (“OR”-connection) and different domains in columns (“AND”-connection). Based on an initial, limited search in the defined databases, additional keywords and domains will be identified. Different terms will be applied for (RQ1) and (RQ2&3), as results of (RQ1) will be used to define keywords for final search regarding (RQ2&3). Reference lists of included articles will be screened for additional literature with relevance for this scoping review (snowball principle) in both steps. The Peer Review of Electronic Search Strategies (PRESS) [38] will be applied to the selection process. The search results will be imported to CITAVI 6 (Swiss Academic Software GmbH, Wädenswil, Switzerland).

### 2.3. Study Selection

The screening of identified records will be performed by three researchers (L.L., L.K. and P.K.), who first independently screen titles and abstracts referring to consent inclusion and exclusion criteria. In case of disagreements, a fourth party (K.-E.C.) will be consulted. After reaching consensus, the same process will be performed for full text screening. RAYYAN Version 2021 (Cambridge, USA) will be used to organize the selection process.

For (RQ1), all references that meet the following inclusion and exclusion criteria (see Table 2) will be considered.

To answer (RQ2&3), articles have to meet the following inclusion and exclusion criteria (see Table 3):

### 2.4. Charting the Data

According to the JBI’S Reviewer Manual [37], relevant studies will be presented in two ways. Regarding (RQ1), author(s), year of publication, country of origin, reference type, OHL definition, conceptual framework and recommendations for future research and practitioners (e.g., human resource managers) will be presented (see Table 4).

For research question (RQ2&3) author(s), year of publication, country of origin, level of evidence, OHL definition, information on intervention as duration and comparator, study population(s), aims of the study, methodology, outcome measures, important results, strengths and weaknesses and recommendations for future research and practitioners will be reported (see Table 5).

L.L., L.K. and P.K. will perform data charting in Microsoft Excel. The first version of data extraction sheets will be piloted by L.L., L.K. and P.K. and adjusted if necessary. L.L. will mainly perform data extraction for included references. L.K. and P.K. will countercheck the work of L.L.

### 2.5. Collating, Summarizing and Reporting the Results

In this last stage, an overview of the reviewed material will be given, categorized if applicable and discussed. The search history will be presented in a flowchart according to the PRISMA 2020 flow diagram [39]. The flowchart will provide information on the number of records identified for each database, register or other sources, the number of records screened and the number of records included. Reasons for exclusion of records will be mentioned.

First, the results of final search and their discussion within the expert-panel regarding (RQ1) will be addresses. Definitions and concepts of OHL in the context of employee health that could be identified will be analyzed and discussed with regard to similarities and differences. A specific focus will lie on the consequences of SARS-CoV-2. The first part will end with a definition of OHL in the context of employee health approved by the expert panel or state why such a definition is not necessary.

Second, results of final search and their discussion within the expert-panel regarding (RQ2&3) will be reported. In a third step the actual state of research on OHL in the context of employee health will be summarized. In case any amendments should be necessary for the procedure described in this protocol, this will be reported and discussed in the final version of the manuscript.

## 3. Discussion

HL is of importance for individuals’ health as low HL is associated with negative health outcomes [3,4,5,6]. The National Action Plan Health Literacy in Germany defines fields of action and recommendations that, i.e., emphasize the promotion of health literacy in all areas of everyday life including occupation and the workplace [1].

The proposed scoping review will contribute to the current stage of research as studies on HL so far mainly focus on OHL in the context of healthcare organizations [14,15,16,17,18,19,20,21,22]. According to the interplay of individual and organizational health literacy [8,31], the aim is to summarize the current state of research on OHL in the context of employee health and to place it in the overall context of HL to give orientation for policy makers, practitioners and researchers. The final version will provide orientation giving information on what matters in the context of OHL and employee health and what specifically needs to be considered, also facing the consequences of SARS-CoV-2.

Especially organizations will benefit from the results of this scoping review, as they gain orientation in the complex field of workplace health promotion (WHP) and corporate health management (CHM). The results may help organizations to improve the quality of WHP measures and CHM on a meta level.

The scoping review will end with a critical discussion on whether a new definition of OHL in the context of employee health is of purpose or if already existing concepts can easily be adapted to the special context of employee health.

### Strengths and Limitations

A major strength of the proposed scoping review is the integration of a multidisciplinary expert-panel following the method of Consensus Development Panels [36].

However, especially in step 1 of the proposed process, some limitations may occur as some forms of grey literature may not easily be identified, e.g., websites and blogs, or may have limited access, e.g., unpublished work manuals. Furthermore, organizations outside the healthcare sector that already focus on OHL or aspects of OHL with regard to employee health may not share public information due to company secret agreements.

## 4. Conclusions

This is, to our knowledge, the first scoping review on organization health literacy in the context of employee health as of yet. Regardless of whether a development of a new definition in this context will be suggested or not, the scoping review will contribute to the current state of research and provide orientation for policy makers, practitioners (e.g., human resource managers) and researchers.

## Figures and Tables

**Table 1 ijerph-19-04381-t001:** PCC framework for identifying relevant studies.

Criteria	Characteristics
Population	- All types of organizations and employers - Employee recipients
Concept	- Definition/conception of OHL - OHL-intervention - Measure(s) of OHL - Recommendations on OHL
Context	- Worldwide - Any organizational setting - Employee health

**Table 2 ijerph-19-04381-t002:** Inclusion and exclusion criteria, RQ1.

Inclusion Criteria	Exclusion Criteria
Articles focusing on OHL in the context of employee health Employee recipients References in German or English language References published at any time Worldwide Any studies, regardless of study design Grey literature E-books or e-book sections	Patient recipients

**Table 3 ijerph-19-04381-t003:** Inclusion and exclusion criteria, RQ2&3.

Inclusion Criteria	Exclusion Criteria
Articles focusing on OHL in the context of employee health Employee recipients References in German or English language References published at any time Worldwide All types of empirical studies	Patient recipients Grey literature Books or book sections

**Table 4 ijerph-19-04381-t004:** Example of data extraction sheet, RQ1.

Author(s)	tbf *
Year of publication	tbf
Country of origin	tbf
Reference type	tbf
OHL definition	tbf
Conceptual framework	tbf
Recommendations for future research	tbf
Recommendations for practitioners	tbf

* tbf = to be filled.

**Table 5 ijerph-19-04381-t005:** Example of data extraction sheet, RQ2&3.

Author(s)	tbf *
Year of publication	tbf
Country of origin	tbf
Level of evidence	tbf
OHL definition	tbf
Study population	tbf
Aim(s) of the study	tbf
Methodology	tbf
Intervention	tbf
Outcome measure(s)	tbf
OHL measure(s)	tbf
Results	tbf
Strengths and weaknesses	tbf
Recommendations for future research	tbf
Recommendations for practitioners	tbf

* tbf = to be filled.

## Data Availability

Not applicable.

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
