# Peer review of "Organizational Health Literacy in the Context of Employee Health: An Expert-Panel-Guided Scoping Review Protocol"

_ijerph, 2022, doi:10.3390/ijerph19074381_

Round 1
Reviewer 1 Report
The paper has merit and is worthy of publication in Open Source. The reviewer agrees that the paper brings forth some ideas that have not be previously considered and provides a solid scoping review.
Author Response
Thank you very much for your support! We carefully checked English language and style.
Reviewer 2 Report
Thank you for the opportunity to read your manuscript, which I believe has significant importance for the special issue "physical and mental health in the workplace" in the field. However, I think this paper is still in a developmental phase and required substantial work. The following specific comments:
1. Abstract
Background: is not clear what main introductions. For instance, you stated "Health literacy (HL) is an interplay of individual and organizational health 9 literacy (OHL)", what the key issues are you provide to present between individual and organisation?
Method: is not clear how many articles identified and included in the study. How did your search time? What data were excluded? What data analysis was performed?
Findings: should be clear what the main results.
Conclusion: should be brief what main summary of the study.
2. Intro
You should provide describe clearly what the main objective of the manuscript in the Introduction and see the conclusions to solve or respond to the research’s questions.
See paragraph 2, page 2 (Organizational health literacy (OHL)...) what you provide to communicate for, paragraph 2, page 2 (For employers, employees’ HL is of special interest as HL can positively influence...) what? Paragraph 3, page 2 (not clear why is both positive?), paragraph 4-7 or line 77-114, page 3 (should be clear what main objectives related to questions).
3. Methods
Your method should be more detailed:
3.1 Protocol (how did you use the scoping review methodological framework). Should be clear why?
3.2 Eligibility criteria (what included and excluded of the studies?) should be more detail why?
3.3 Information sources and search strategy (how did you search?)
3.4 Study selection process (how did your select studies?)
3.5 Data items and data collection process
3.6 Methodological quality appraisal
3.7 Synthesis
Or see the sample of scoping review: https://link.springer.com/content/pdf/10.1186/s12874-016-0116-4.pdf
4. Findings
Your findings should be clearly presented. What main findings? I found only few evidences support your study. The findings are invalid review supports. Should be reviewed clear what/why/how organizational health literacy matters? What relates to employees' health? What new contribution to exist current knowledge in organizational health literacy?
* Should be clear above mention.
5. Discussion
Your discussion is too limited and mishandling debate/discussion with the main findings. Should be revised all paragraphs, because are not applicable for discussion.
*Strengths and Limitations
I have not found what strength of the study. Should be clear what/why? This is because you are writing too limitation structure, only 7 sentences, because of the reviewers did not understand what the limitations of the study.
6. Conclusion
Your conclusion is too limited. Why did not follow with the main findings. I found only 5 sentences. What main findings with summarization?
7. References
Both text and list references are incorrect. Should be up-to-date what new article related to your study.
Author Response
Thank you for revising our manuscript and your comments, which we will answer point-by-point in the following:
COMMENTS: Background: is not clear what main introductions. For instance, you stated "Health literacy (HL) is an interplay of individual and organizational health 9 literacy (OHL)", what the key issues are you provide to present between individual and organisation?
ANSWER: Thank you for your comments. To clarify, we added key issues of individual and organizational health literacy. It now says:
“In a holistic view, HL demonstrates an interplay of individual (individual skills and abilities) and organizational (system demands and complexities) HL. Low HL does not exclusively result from a lack of knowledge, lack of motivation or insufficient skills of the individual but is also decisively shaped by the societal, lifeworld and social conditions under which individuals live and the demands of life situations and environ-mental aspects [8,30,31].” (see lines 88 to 93)
Furthermore we added
“In Germany, employers are subject to the Occupational Health and Safety Act and have a duty to detect and prevent or mitigate potential health hazards for employers. Furthermore, in the event of longer periods of absence, they are also obliged to offer employees support in form of an occupational integration management [29]. It is not only up to employees to take care of health issues, but also employers bear responsibility.” (see lines 82 to 86)
COMMENTS: Method: is not clear how many articles identified and included in the study. How did your search time? What data were excluded? What data analysis was performed?
Findings: should be clear what the main results.
Conclusion: should be brief what main summary of the study.
ANSWER: As this is a protocol, we will consider this aspects in the final version of the scoping review.
COMMENTS:
- Intro
You should provide describe clearly what the main objective of the manuscript in the Introduction and see the conclusions to solve or respond to the research’s questions.
ANSWER: The objective of this protocol is to describe the aims of a proposed scoping review, that will be guided by an expert-panel. Objectives are described in the introduction in lines 123 to 146. The objectives of the proposed scoping review and added value of the study is also mentioned in the discussion, please see lines 256 to 268.
COMMENTS: See paragraph 2, page 2 (Organizational health literacy (OHL)...) what you provide to communicate for, paragraph 2, page 2 (For employers, employees’ HL is of special interest as HL can positively influence...) what? Paragraph 3, page 2 (not clear why is both positive?), paragraph 4-7 or line 77-114, page 3 (should be clear what main objectives related to questions).
ANSWER: Thank you for your comments. We added some examples to clarify what kind of measures OHL may help individuals to support individuals to take care of their health. It now says: “For example, digital applications open up the opportunity to facilitate access to the health system and to health care services and also to lower time-related usage thresh-olds [8]. Especially in the continuum between prevention and rehabilitation, HL is of special interest, as HL can contribute to prevent or slowing down deterioration of the disease situation and strengthening health resources [10–13].” (see lines 59 to 64)
Furthermore, re rephrased the sentence “For employers, employees’ HL is of special interest as HL can positively influence...”. It now says: “Employees’ HL is of special interest also for employers: HL can positively influence individual skills and work ability and reduce occupational hazards and injuries. Employees’ work ability ensures organizations’ competitiveness and overall productivity [23–26].”, see line 71 to 73.
COMMENTS:
Paragraph 3, page 2 (not clear why is both positive?): Here, we want to express that there are positive as well as negative effects. (see now line 98)
paragraph 4-7 or line 77-114, page 3 (should be clear what main objectives related to questions:
ANSWER: The objective of research question 1 is to identify concepts and definitions of organizational health literacy that focus on any aspects of employee health in a first search and discuss results within the planned expert-panel. Based on the results of the expert panel, research question 2 then aims at identifying empirical studies that focus on any aspects of organizational health literacy (OHL) with the aim to improve employee health. Furthermore, we will examine which and whether concepts and definitions of OHL were used. Research question 3 aims at identifying measures that were used to operationalize organizational health literacy within the studies, see line 139 to 146. As the objectives, in our opinion, directly emerge from research questions, we would not elaborate further at this point.
COMMENTS:
- Methods
Your method should be more detailed:
3.1 Protocol (how did you use the scoping review methodological framework). Should be clear why?
3.2 Eligibility criteria (what included and excluded of the studies?) should be more detail why?
3.3 Information sources and search strategy (how did you search?)
3.4 Study selection process (how did your select studies?)
3.5 Data items and data collection process
3.6 Methodological quality appraisal
3.7 Synthesis
Or see the sample of scoping review: https://link.springer.com/content/pdf/10.1186/s12874-016-0116-4.pdf
ANSWER: Thank you for your comments. As this is just the protocol, we will consider those comments in the final version of the scoping review.
COMMENTS:
- Findings
Your findings should be clearly presented. What main findings? I found only few evidences support your study. The findings are invalid review supports. Should be reviewed clear what/why/how organizational health literacy matters? What relates to employees' health? What new contribution to exist current knowledge in organizational health literacy?
* Should be clear above mention.
- Discussion
Your discussion is too limited and mishandling debate/discussion with the main findings. Should be revised all paragraphs, because are not applicable for discussion.
ANSWER: As this is a protocol, we are not able to provide and discuss final results of the process.
COMMENTS:
*Strengths and Limitations
I have not found what strength of the study. Should be clear what/why? This is because you are writing too limitation structure, only 7 sentences, because of the reviewers did not understand what the limitations of the study.
ANSWER: At this stage of the process, we are not able to discuss final strengths and limitations of the scoping review, as more strengths and limitations may emerge from final search and expert-panel discussion.
COMMENTS:
- Conclusion
Your conclusion is too limited. Why did not follow with the main findings. I found only 5 sentences. What main findings with summarization?
ANSWER: As this is a protocol, we are not able to follow main findings in our conclusion.
COMMENTS:
- References
Both text and list references are incorrect. Should be up-to-date what new article related to your study.
ANSWER: Thank you for that comment. We added some more references to our study and revised and checked all references.
Reviewer 3 Report
Dear Authors,
Thank you for offering me the opportunity to review this paper.
The theme is undoubtedly exciting, the methodology well described, but the presentation of the results is entirely lacking.
The literature analysis must be returned to the reader. Otherwise, the usefulness of the research and its future replicability is not understood.
The paper should certainly be expanded by presenting the most exciting or relevant works and topics that the existing literature has treated.
Moreover, in the introduction, it is necessary to explain better the HLS19 Consortium of the WHO Action Network M-POHL.
Furthermore, it seems that:
i) none of the research questions have been answered
ii) the questions are not consistent with the development of the article and the research nor with the title.
The validation of the process, although elaborated by experts, I am sure, must be carried out based on the results returned, not on the mere description of the methodology.
In conclusion, at the moment, the paper is incomplete and should be integrated with a more detailed presentation of the results emerging from the analysis of the literature. It could be helpful to present them by themes or topics, reclassifying the literature based on a more rigorous theoretical framework. Otherwise, it seems that the conclusions (synthetic, actually) cannot be generalizable or replicable.
Good luck with your research!
Author Response
Thank you very much for revising our manuscript. We will answer your comments point-by-point in the following:
COMMENTS:
The theme is undoubtedly exciting, the methodology well described, but the presentation of the results is entirely lacking.
The literature analysis must be returned to the reader. Otherwise, the usefulness of the research and its future replicability is not understood.
The paper should certainly be expanded by presenting the most exciting or relevant works and topics that the existing literature has treated.
ANSWER: Thank you for your comments. We added some more references to the protocol. More relevant literature will be identified within the process following this protocol for the final version of the scoping review. As this is a protocol, we are not able to present results at this stage.
COMMENTS:
Moreover, in the introduction, it is necessary to explain better the HLS19 Consortium of the WHO Action Network M-POHL.
Furthermore, it seems that:
i) none of the research questions have been answered
ii) the questions are not consistent with the development of the article and the research nor with the title.
ANSWER: Thank you for your comment. We added an explanation on the HLS19 Consortium of the WHO Action Network M-POHL. It now says: “The World Health Organization (WHO)/Europe initiated the Action Network on Measuring Population and Organizational Health Literacy (M-POHL) to regularly measure HL. The first survey was carried out in 17 countries (Austria, Belgium, Bulgaria, Czech Republic, Denmark, France, Germany, Hungary, Ireland, Israel, Italy, Norway, Portugal, Russian Federation, Slovakia, Slovenia, and Switzerland) in 2019 (HLS19) [1,2].”, see line 47 to 52.
i) As this is a protocol, we will answer the research questions in the final version of the scoping review.
ii) Could you please clarify what you mean here? We choose the title as it should clarify that this manuscript is a protocol for an expert-panel guided scoping review on OHL in the context of employee health. Our aim is that the final version of the scoping review answers the three research questions on OHL in the context of employee health.
COMMENTS:
The validation of the process, although elaborated by experts, I am sure, must be carried out based on the results returned, not on the mere description of the methodology.
In conclusion, at the moment, the paper is incomplete and should be integrated with a more detailed presentation of the results emerging from the analysis of the literature. It could be helpful to present them by themes or topics, reclassifying the literature based on a more rigorous theoretical framework. Otherwise, it seems that the conclusions (synthetic, actually) cannot be generalizable or replicable.
ANSWER: Thank you for your comments. As we have no results yet, we will consider your comments in the manuscript of the final scoping review.
Reviewer 4 Report
Readers will be interested above all at a later stage, ie in reading the results of the scoping review; in particular, policy makers and practicioners will be interested in reading the results of the step 2 of the research from which they will be able to draw useful information to support their decisions.
Author Response
Thank you very much for your supporting revision! We carefully checked English language and style.
Reviewer 5 Report
This protocol is very interesting and the manuscript is generally well written. Methods and explanations are quite good. I believe it can be accepted in current form.
Author Response
Thank you very much for your supportive revision! We carefully checked English language and style.
Reviewer 6 Report
In my opinion, this represents a very interesting topic, the manuscript however shows some weaknesses which require a further review and a careful revision of the language through the text. However, the captions in Tables should be amended. In addition, English is decent but I suggest a thorough review of the manuscript before accepting it for publication. To further improve the text, I suggest the following changes in the manuscript.
Abstract: Abstract should be written in concise. I would suggest listing only some of the most important results to justify the implications and conclusions of the study.
The background of an introduction should be revised accordingly.
The introduction is very good. It doesn't reflect the goal; please rewrite it again, it is suggested to include some latest reference.
Objectives of this study must be included at end of introduction part.
I highly recommended to authors, if possible, please modify the figure with good quality images.
The economic intuition behind the results are missing. The author/s should revise the discussion part. The result should be supported with recent studies.
What is contribution of this work to existing literature?
It has been observed that the authors have used old references and ignored the latest studies. So it is suggested to add recent references. Please check reference section some references are missing.The policy implications also required elaboration. The implications should go along with the results and the course of action should be discussed in this part.In some places, some grammatical errors are found that need to be fixed.
Author Response
Thank you very much for revising our manuscript! We will answer your comments point-by-point in the following:
COMMENTS:
Abstract: Abstract should be written in concise. I would suggest listing only some of the most important results to justify the implications and conclusions of the study.
ANSWER: Thank you for your comment. We revised the abstract, that is now written more concisely. As this is a protocol, we are not able to present results at this stage of the process.
COMMENTS:
The background of an introduction should be revised accordingly. The introduction is very good. It doesn't reflect the goal; please rewrite it again, it is suggested to include some latest reference. Objectives of this study must be included at end of introduction part.
ANSWER: Thank you for your comment. We revised the introduction and added information and references.
It now says:
“Especially in the continuum between prevention and rehabilitation, HL is of special interest, as HL can contribute to prevent or slowing down deterioration of the disease situation and strengthening health resources [10–13].” (see lines 61 to 64)
“The reduction of occupational hazards, injuries and absenteeism are an indication that HL also has an effect on economic costs (, e.g. fewer absence days in the company due to incapacity to work). In a scoping review on costs of limited HL, Eichler et al. [27] state that the costs of limited HL are potentially significant. In 2009, limited HL was assumed to cause 3-5% of the total health care budget in Canada [28].” (see lines 76 to 81)
“In Germany, employers are subject to the Occupational Health and Safety Act and have a duty to detect and prevent or mitigate potential health hazards for employers. Furthermore, in the event of longer periods of absence, they are also obliged to offer employees support in form of an occupational integration management [29]. It is not only up to employees to take care of health issues, but also employers bear responsibility.” (see lines 82 to 86)
“Low HL does not exclusively result from a lack of knowledge, lack of motivation or insufficient skills of the individual but is also decisively shaped by the societal, life-world and social conditions under which individuals live and the demands of life situations and environmental aspects [8,30,31].” (see lines 90 to 93)
“However, low earners and employees with low skill levels are less likely to be able to work from home. As employees without home office access bear a disproportionately higher burden of the pandemic (higher home office potential is negatively correlated with both regional reports of short-time work and regional infection figures) [34] and as financial deprivation and self-perceived level in society predict general and specific HL [2], those differences in individuals with and without home office possibilities have to be considered. Results will be discussed with regard to the requirements this entails.” (see lines 101 to 107)
COMMENTS: Objectives of this study must be included at end of introduction part.
ANSWER: We stated the objectives of the study at the end of the introduction part, see lines 123 to 146. If you meant that something else is missing, could you please clarify?
COMMENTS: I highly recommended to authors, if possible, please modify the figure with good quality images.
ANSWER: In this protocol we only used tables. There appeared to be some issues with the format during upload which we revised.
COMMENTS: The economic intuition behind the results are missing. The author/s should revise the discussion part. The result should be supported with recent studies. What is contribution of this work to existing literature?
ANSWER: Thank you very much for your comment. We added some information. It now says: “The reduction of occupational hazards, injuries and absenteeism are an indication that HL also has an effect on economic costs (, e.g. fewer absence days in the company due to incapacity to work). In a scoping review on costs of limited HL, Eichler et al. [27] state that the costs of limited HL are potentially significant. In 2009, limited HL was assumed to cause 3-5% of the total health care budget in Canada [28].” (see lines 76 to 81) As we have no results yet, we are not able to include them in our discussion section.
COMMENTS: It has been observed that the authors have used old references and ignored the latest studies. So it is suggested to add recent references. Please check reference section some references are missing.
ANSWER: Thank you for that comments. We added more information and references in the introduction part. Additionally, we are looking forward to identify more relevant studies in the described process of the proposed scoping review. We revised the reference section to hopefully solve the problem with missing references.
COMMENTS: The policy implications also required elaboration. The implications should go along with the results and the course of action should be discussed in this part.
ANSWER: Thank you for your comment. As this is a protocol, at this stage, we are not able to draw concrete implications from results. We will do this in the final version of the scoping review.
COMMENTS: In some places, some grammatical errors are found that need to be fixed.
ANSWER: Thank you for your hint. We carefully checked English language and style.
Round 2
Reviewer 2 Report
Thank you for your revision. The revised version does not make any changes from the earlier version. But, some points clearly respond to reviewers about the limitation of the scope review. I accepted with moderate contributions to a special issue of "Physical and Mental Health in the Workplace". However, this review seems to contribute to existing new knowledge implications.
Author Response
Thank you very much for revising our protocol again and suggesting the revised version for publication. We revised the original version of the manuscript in round 1 and used the “Track Changes” function in MS Word. We also revised English language and style. To our opinion, English language and style follow the current grammatical rules and uses the terminology appropriate to subject area. Could you please clarify if your comments referred to something else or were misunderstood by us in this case?
We appreciate your time and effort for the revision and will consider your comments when composing the final version of the scoping review!
Reviewer 3 Report
Dear Authors,
I very much appreciated the revisions you made to the paper.
Unfortunately, my doubts still remain regarding the construction of the protocol.
The research questions are straightforward, but your "protocol" about what factors to include/exclude, the determining factors, etc., must be taken as absolute truth.
According to the paper's construction, the conclusion (the protocol) must be assumed by the reader as a datum or an assumption of unlimited validity.
You indicate the selection methodology of the existing studies but do not indicate the studies analyzed nor the methods used for their analysis (e.g. content analysis or other).
What continues to leave me very perplexed is the validity of the conclusions.
I suggest that the authors prepare a summary table that includes at least: i) the studies analyzed, ii) the fundamental concepts identified, iii) the methodology used to analyze the relevant literature.
Otherwise, the "credibility" of the determinants and conclusions cannot be accepted.
Good luck with your work!
Author Response
Thank you very much, again, for your comments! We will provide point-by-point answers in the following:
Comment: The research questions are straightforward, but your "protocol" about what factors to include/exclude, the determining factors, etc., must be taken as absolute truth.
Answer: Thank you for your comment. At this stage of the process, we find it difficult to formulate final including/excluding factors, as (1) the final search strategy of the literature search (step two in the process) depends on the results discussed in the expert-panel, and (2) an initial, limited search in the defined databases will precede the final search strategy.
Comment: According to the paper's construction, the conclusion (the protocol) must be assumed by the reader as a datum or an assumption of unlimited validity.
Answer: Unfortunately, we are not sure if we understand you correctly here. With “yet” in the first sentence of the conclusion we want to clarify, that the proposed scoping review will be, to our knowledge, (yet,) the first scoping review on organization health literacy in the context of employee health. Does this answer your comment or could you please clarify if necessary?
Comment: You indicate the selection methodology of the existing studies but do not indicate the studies analyzed nor the methods used for their analysis (e.g. content analysis or other).
Answer: As this is a protocol, we were, at this point, not able to analyze studies. We provide information on how we plan to present results of the included studies in table 4 and table 5.
Comment: What continues to leave me very perplexed is the validity of the conclusions.
Answer: Could you clarify what you mean here? We are convinced that the results of the proposed scoping review will contribute to the current state of research by providing orientation for policy makers, practitioners (e.g. human resource managers) and researchers, regardless of review results.
Comment: I suggest that the authors prepare a summary table that includes at least: i) the studies analyzed, ii) the fundamental concepts identified, iii) the methodology used to analyze the relevant literature. Otherwise, the "credibility" of the determinants and conclusions cannot be accepted.
Answer: Thank you for this comment. We will consider this in the final version of the scoping review. As this is just the protocol describing our future project, we are not able to provide information on e.g. analyzed studies or identified concepts.
We appreciate your time and effort for the revision and will consider your comments when composing the final version of the scoping review!
Reviewer 6 Report
-
Author Response
Thank you very much for revising our manuscript again. We appreciate your time and effort for the revision and will consider your comments (of both review rounds) when composing the final version of the scoping review!